# MULTICRAFTER: HIGH-FIDELITY MULTI-SUBJECT GENERATION VIA DISENTANGLED ATTENTION AND IDENTITY-AWARE PREFERENCE ALIGNMENT

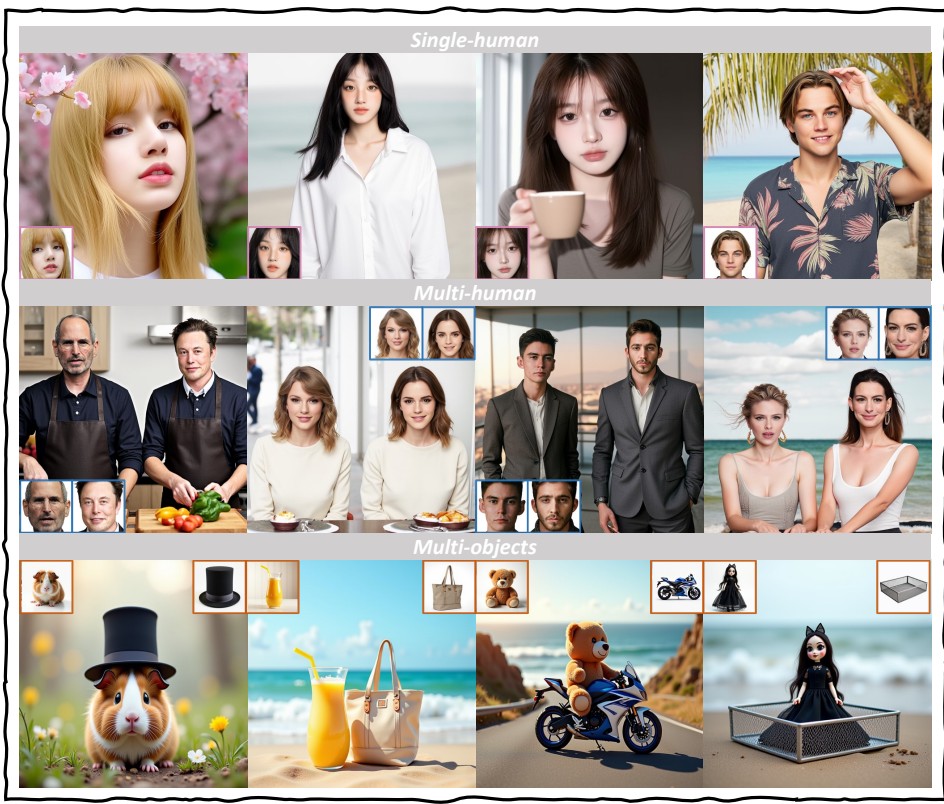

Figure 1: MultiCrafter enables multi-subject personalization. Input are surrounded by squares.

## ABSTRACT

Multi-subject image generation aims to synthesize user-provided subjects in a single image while preserving subject fidelity, ensuring prompt consistency, and aligning with human aesthetic preferences. However, existing methods, particularly those built on the In-Context-Learning paradigm, are limited by their reliance on simple reconstruction-based objectives, leading to both severe attribute leakage that compromises subject fidelity and failing to align with nuanced human preferences. To address this, we propose MultiCrafter, a framework that ensures high-fidelity, preference-aligned generation. First, we find that the root cause of attribute leakage is a significant entanglement of attention between different subjects during the generation process. Therefore, we introduce explicit positional supervision to explicitly separate attention regions for each subject, effectively mitigating attribute leakage. To enable the model to accurately plan the attention region of different subjects in diverse scenarios, we employ a Mixture-of-Experts architecture to enhance the model's capacity, allowing different experts to focus on different scenarios. Finally, we design a novel online reinforcement learning framework to align the model with human preferences, featuring a scoring mechanism to accurately assess multi-subject fidelity and a more stable training strategy tailored for the MoE architecture. Experiments validate that our framework significantly improves subject fidelity while aligning with human preferences better.

# 1 INTRODUCTION

Subject-driven image generation, which aims to create images featuring user-provided subjects, has become a cornerstone of personalized content creation. Propelled by higher-quality data and the widespread adoption of Diffusion Transformers (Labs, 2024; Esser et al., 2024; Gao et al., 2025; Li et al., 2024b), text-to-image models have seen rapid advancements. This progress has significantly enhanced single-subject image generation, where models now excel at preserving subject fidelity (Li et al., 2025c; Feng et al., 2025; He et al., 2025). Among the various techniques, the In-Context-Learning (ICL) paradigm (Huang et al., 2024a;b; Tan et al., 2024) has emerged as a mainstream approach. Its high adaptability to the transformer architecture and strong capability for maintaining subject fidelity have made it particularly effective for single-subject tasks.

However, extending this success from single-subject to multi-subject generation is not trivial. Building on the ICL framework, recent works like UNO (Wu et al., 2025c) and OmniGen (Xiao et al., 2025) have ventured into this complex multi-subject customization task. Despite their efforts, these methods often produce suboptimal results, especially when dealing with intricate subjects like human faces. They frequently suffer from suboptimal results, such as identity fusion and attribute leakage between subjects. This raises a critical question that forms the very foundation of our work: **why do existing ICL-based methods falter in the multi-subject setting?**

The primary challenge stems from the training objective. Existing ICL-based methods are typically optimized with a simple reconstruction loss. This objective implicitly tasks the model with the dual responsibilities of distinguishing subject features and arranging them spatially. However, such supervision alone proves insufficient for the complexities of multi-subject scenarios. As shown in Fig. 2, this inadequacy leads to an undesired entanglement between subject-specific attention fields in these methods, like UNO. This phenomenon, which we term attention bleeding, causes attribute leakage and severely damages subject fidelity. Furthermore, a simple reconstruction objective fails to capture nuanced human preferences, such as aesthetic quality and precise prompt alignment.

To address these limitations, we introduce MultiCrafter, a framework that achieves high-fidelity, preference-aligned multi-subject image generation through three key innovations. To address the attribute leakage caused by attention bleeding, we propose an **Identity-Disentangled Attention Regularization**. This mechanism applies explicit positional supervision only during the training phase to double blocks in FLUX (Labs, 2024), which are pivotal regions for feature injection and spatial control. This compels the model to distinguish between different subject features and learn distinct, disentanglement attention regions for each subject, drastically reducing attribute leakage.

Considering a single model struggles to cope with various attention layouts caused by a diverse range of subjects and prompts, we enhance its capacity by incorporating a Mixture-of-Experts (MoE) architecture. Inspired by MoE-LoRA's success in multitask tuning (Feng et al., 2024; Zhang et al., 2025), our **Efficient Adaptive Expert Tuning** allows different expert networks to specialize in varied scenarios, dynamically selected via a routing mechanism. This ensure our method to maintain excellent subject fidelity in various scenarios without an increase in inference complexity.

For aligning with human aesthetic and semantic preferences while ensuring subject fidelity, we design a novel online reinforcement learning framework. We introduce **Identity-Preserving Preference Optimization** that aligns the model across three axes: aesthetic quality, text-image alignment, and subject fidelity. To accurately measure subject fidelity, we introduce Multi-ID Alignment Reward, which use the Hungarian matching algorithm to maximize the overall match quality between multiple generated subjects and their references for precise scoring. Besides, we introduce Group Sequence Policy Optimization (GSPO) Zheng et al. (2025) to adapt to our MoE-LoRA design, thereby avoiding the core issue of training instability caused by expert-activation volatility inherent in MoE architectures. Our experiments demonstrate that MultiCrafter achieves significant improvements over existing methods. Our main contributions are as follows:

- We propose explicit positional supervision that disentangles attention across subjects, thereby reducing attribute leakage and enhancing subject fidelity.
- We integrate an MoE-LoRA architecture that increases capacity for diverse subjects and spatial layouts while maintaining inference efficiency.
- We design the first online reinforcement learning framework tailored for multi-subject generation, introducing a Multi-ID Alignment Reward and GSPO for stable and preference-aligned training.

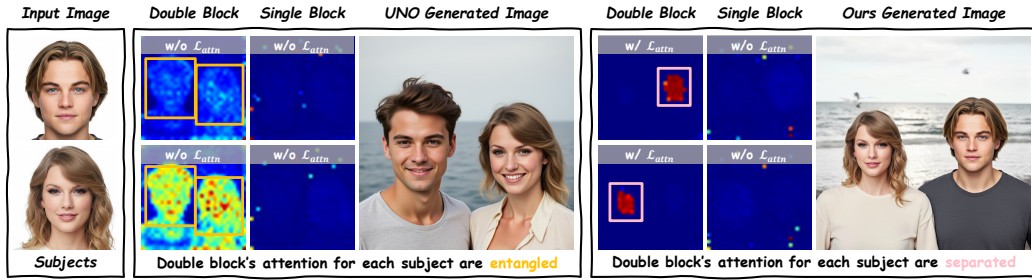

Figure 2: Visual comparison of attention maps. The ICL-based method UNO, which only relies on reconstruction loss (left), fails to preserve the subject fidelity, where the double block's attention regions for each subject are entangled, leading to attribute leakage. Our method overcomes this problem and maintains subject fidelity.

## 2 RELATED WORK

**Subject-driven Generation** has attracted increasing attention (Chen et al., 2023; Han et al., 2023; Shi et al., 2024; Ruiz et al., 2024; Hua et al., 2023; Han et al., 2024; Liu et al., 2023a;b; Gu et al., 2024; Feng et al., 2025). These works can be broadly categorized into two types. (1) Fine-tuning-based methods, includes methods such as Textual Inversion (Gal et al., 2022), DreamBooth (Ruiz et al., 2023), and Custom Diffusion (Kumari et al., 2023). These approaches achieve customization by fine-tuning part of the model's parameters. (2) Tuning-free methods, like IP-Adapter (Ye et al., 2023), InstantID (Wang et al., 2024), and PhotoMaker (Li et al., 2024a). These approaches leverage large-scale training to eliminate the need for retraining when the subject changes. However, customized generation of multiple subjects from a single image introduces new challenges, especially in maintaining individual subject fidelity and mitigating attribute entanglement. Recently, the powerful capabilities of foundation models based on the DiT (Peebles & Xie, 2023) architecture have greatly enhanced the generation of multiple subjects, leading to the emergence of a series of works such as HunyuanCustom (Hu et al., 2025), OmniControl (Tan et al., 2024), UniReal (Chen et al., 2025b), UNO (Wu et al., 2025c), DreamO (Mou et al., 2025) and XVerse (Chen et al., 2025a).

**Reinforcement Learning for Text-to-Image Generation** has become an active area of research. Initial strategies included policy gradient methods like Proximal Policy Optimization (PPO) (Schulman et al., 2017; Black et al., 2023; Fan et al., 2023; Gupta et al., 2025; Miao et al., 2024; Zhao et al., 2025). A subsequent major development is the adoption of Direct Preference Optimization (DPO) and its variants (Wallace et al., 2024; Yang et al., 2024; Yuan et al., 2024; Liu et al., 2025b; Zhang et al., 2024; Furuta et al., 2024; Li et al., 2025a). Some recent works have introduced online RL technology, Group Relative Policy Optimization (GRPO) Shao et al. (2024), into Text-to-Image Generation, achieving significant performance gains. Flow-GRPO (Liu et al., 2025a), DanceGRPO (Xue et al., 2025) introduce exploration by reformulating the deterministic Ordinary Differential Equation (ODE) of flow-matching models into a Stochastic Differential Equation (SDE). The improved MixGRPO (Li et al., 2025b) further boosts training efficiency with a mixed ODE-SDE framework. However, these methods have been limited to the basic text-to-image task, leaving the application of online reinforcement learning to multi-subject driven generation largely unexplored.

## 3 PRELIMINARY

**Flow Matching**. Flow Matching (Lipman et al., 2022) is gradually replacing DDPM as the mainstream for text-to-image models due to its more efficient sampling strategies. These models typically first train an autoencoder (consisting of an encoder $\mathcal{E}$ and a decoder $\mathcal{D}$) to obtain the latent space representation $z_0 = \mathcal{E}(x)$ of an image $x$. Let $z_0$ be a data sample, $\epsilon \in \mathcal{N}(0,1)$ is the Gaussian noise, and $c_{text}$ be the prompt of this image. Flow Matching formulates generation as a continuous transformation along an Ordinary Differential Equation (ODE), $\frac{dz_t}{dt} = v(z_t, t), \ t \in [0,1]$, which deterministically maps noise to data. The interpolated data at time $t$ is $z_t = (1-t)z_0 + t\epsilon$. A neural network $v_\theta(x_t, t)$ is trained to approximate the velocity field of this ODE, with the objective

$$\mathcal{L}_{diff} = \mathbb{E}_{t,z_0,\epsilon \in \mathcal{N}(0,1)} \|v - v_\theta(z_t, t, c_{text})\|^2, \quad v = z_0 - \epsilon. \tag{1}$$

The DiT framework and Flow Matching are widely used in recent diffusion models, such as Stable Diffusion3 (Esser et al., 2024) and Flux (Labs, 2024). In this paper, we use Flux as our base model.

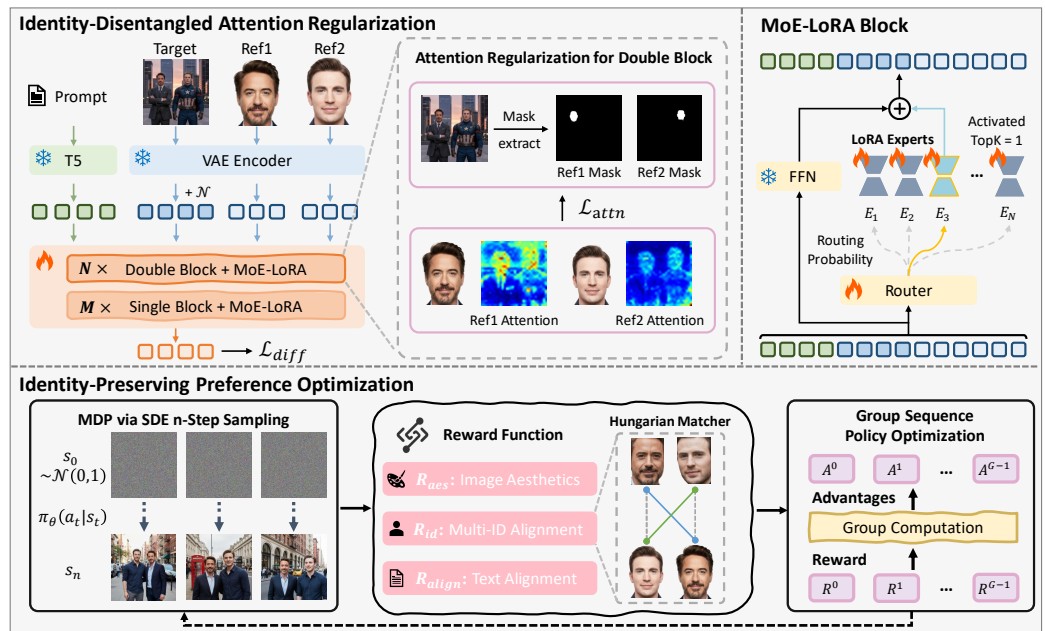

Figure 3: Overall pipeline of MultiCrafter. Our framework is built on three core innovations: (Top Left) Identity-Disentangled Attention Regularization uses positional supervision to prevent attribute leakage; (Top Right) the MoE-LORA architecture boosts model capacity for diverse scenarios; and (Bottom) the Identity-Preserving Preference Alignment framework employs a novel online reinforcement learning strategy with a Multi-ID Alignment Reward and the stable GSPO algorithm to align the model with human preferences.

**Group Relative Policy Optimization (GRPO).** GRPO struggles with ODE-based flow models because their deterministic systems lack the inherent stochasticity essential for reinforcement learning frameworks. So DanceGRPO Xue et al. (2025) and Flow-GRPO Liu et al. (2025a) convert ODEs to SDEs, enabling stochastic reinforcement learning for image generation. MixGRPO Li et al. (2025b) further improves efficiency by applying this only within a sliding time window during training. Given the prompt $c_{text}$, the training process in MixGRPO is similar to Flow-GRPO and DanceGRPO, but only optimizes the time steps sampled within the interval $S$. The behavior of this window is governed by key hyperparameters: the window size $w$ sets the number of consecutive timesteps to optimize at once, the shift interval $\tau$ determines how many training iterations pass before the window moves, and the window stride $s$ specifies how many timesteps the window advances during a shift. The final training objective is given by:

$$J(\theta) = \mathbb{E}_{c \sim c_{text}, \{x_i^T\}_{i=0}^N \sim \pi_{\theta_{old}}(\cdot|c)} \left[ \frac{1}{N} \sum_{i=1}^{N} \frac{1}{|S|} \sum_{t \in S} \min \left( r_i^t(\theta) A_i, clip(r_i^t(\theta), 1 - \beta, 1 + \beta) A_i \right) \right],$$
(2)

where $r_i^t(\theta)$ is the policy ratio and $A_i$ is the advantage score. $\beta$ is a hyperparameter that serves to clip the policy ratio, ensuring stable updates, and

$$r_i^t(\theta) = \frac{\pi_\theta(x_{t+1} \mid x_t, c)}{\pi_{\theta_{old}}(x_{t+1} \mid x_t, c)}, \quad A_i = \frac{R(x_i^T, c) - mean\{R(x_i^T, c)\}_{i=1}^N}{std\{R(x_i^T, c)\}_{i=1}^N},$$
(3)

where $\pi_\theta(x_{t+1} \mid x_t, c)$ is the policy function and $R(x_i^0, c)$ is provided by the reward model.

## 4 METHODS

In this section, we first explore the underlying causes of feature entanglement in multi-subject customized generation and present the overall framework of our method in Sec. 4.1. Then, in Sec. 4.2, we introduce **Identity-Disentangled Attention Regularization** to address attribute leakage issues arising from attention bleeding. To enhance the model's ability to maintain subject fidelity in different scenarios, we introduce **Efficient Adaptive Expert Tuning** in Sec. 4.3. Lastly, we incorporate **Identity-Preserving Preference Optimization** in Sec. 4.4, leveraging reinforcement learning to align the model with human preferences.

### 4.1 EXPLORE THE IN-COTEXT-LEARNING

Given $N$ reference images of different subjects, we aim to enable the model to generate images of these subjects according to a text prompt. Existing ICL-based methods encodes the $N$ subject images through an encoder $\mathcal{E}$ to obtain latent space feature of these subjects, $\mathcal{Z} = \{\mathbf{z}_i^{ref}\}_{i=1}^N$, where each $\mathbf{z}_i^{ref} \in \mathbb{R}^{c \times w \times h}$, $(w, h)$ is spatial size, $c$ is the number of channels. However, we find that when multiple subjects are input, especially when the attributes of the subjects themselves are similar, it is easy to cause confusion between the attributes of different subjects, resulting in a decrease in subject fidelity. To investigate the underlying reasons, we leveraged a representative method (Wu et al., 2025c) to visualize the attention maps between each reference feature $\mathbf{z}_i^{ref}$ and the feature of the generated image $\hat{z}$. Since the base model Flux comprises both double block and single block structures, we visualized the attention maps for each block type separately.

As shown on the left of Fig. 2, we identify two key phenomena. First, the double blocks of Flux are far more pivotal in determining the spatial layout of the reference subject than the single block. Second, methods trained solely on a reconstruction loss lead to an undesired entanglement between subject-specific attention fields, causing attribute leakage and severely compromising subject fidelity. We can therefore infer a critical condition for high fidelity: the peak response within the double block's attention scores, corresponding to a specific subject, must consistently align with that subject's spatial region in the generated output. We leverage this in our method. Furthermore, this simple supervision method fails to account for human preferences adequately. Therefore, we introduce reinforcement learning as a post-training to align human preferences.

### 4.2 IDENTITY-DISENTANGLED ATTENTION REGULARIZATION

Based on that, the subject fidelity can be enhanced by strictly aligning the hot spot areas of the attention scores of double blocks with the spatial positions of the corresponding subjects in the generated image. We design a simple but effective regularization. Specifically, during the training process, for the latent space feature $\mathbf{z}_i^{ref}$ of the $i$-th reference subject. Following (Wu et al., 2025c), we partition the reference feature $\mathbf{z}_i^{ref}$ into patches and apply positional encoding, resulting in a sequence of 1D tokens $\mathbf{z}_i^{r\prime} \in \mathbb{R}^{l \times c}$, $l$ is the number of tokens, $c$ is the number of channels. Then, we can obtain its attention map with the generated content $z_t$ at the $k$-th double block:

$$m_k^i = \text{Softmax}(\frac{\mathbf{Q}_{k,i}\mathbf{K}_k^T}{\sqrt{d}}), \tag{4}$$

where $\mathbf{Q}_{k,i} \in \mathbb{R}^{l \times c}$ is the query generated from the $i$-th subject image within the $k$-th double block, $\mathbf{K} \in \mathbb{R}^{l_t \times c}$ is the key produced by the noisy image latent tokens in the current layer, and $l_t$ denotes the number of tokens in the noisy image latent. For a model with $K$ double blocks, we can obtain the attention maps corresponding to the $i$-th subject from all blocks, which are aggregated into a set $\{m_1^i, m_2^i, ..., m_K^i\}$. We then average and normalize this set to obtain the mean attention map $\hat{M}_i$. By pre-annotating the training data, we obtain the ground-truth mask $M_i$ corresponding to the $i$-th subject within the generated image. Notably, for human subjects, we exclusively use the facial region as the reference image and similarly focus only on the facial area for the generated image.

Finally, we employ the dice loss (Milletari et al., 2016), a standard loss function for segmentation tasks, to minimize the discrepancy between each ground-truth mask $M_i$ and the corresponding mean attention map $\hat{M}_i$. The formulation is as follows:

$$\mathcal{L}_{attn} = \sum_{i=1}^N \left( 1 - \frac{2\sum_j (\hat{M}_{i,j} \cdot M_{i,j}) + \epsilon}{\sum_j \hat{M}_{i,j} + \sum_j M_{i,j} + \epsilon} \right), \tag{5}$$

where the index $j$ iterates over all spatial locations of the maps, and $\epsilon$ is a small constant added for numerical stability. By minimizing this attention regularization loss $\mathcal{L}_{attn}$, we explicitly encourage the model's attention mechanism to concentrate on the precise spatial regions occupied by each subject. This forces a spatial disentanglement of subjects within the attention maps. This avoids attribute leakage and improves the subject fidelity of the generated image. The final loss function of our framework is defined as ($\lambda$ is a factor that balances the loss weight):

$$L = L_{diff} + \lambda \cdot L_{attn}. \tag{6}$$

### 4.3 EFFICIENT ADAPTIVE EXPERT TUNING

For efficient training, existing ICL-based methods usually use LoRA to fine-tune the model. But the significant variance in the spatial layout of subjects across various subjects and prompts poses a challenge for a standard LoRA, whose limited capacity is insufficient for diverse subject-driven generation. To address this, and inspired by the success of MoE-LoRA in multitask tuning (Feng et al., 2024; Liu et al., 2024; Gou et al., 2023), we adopt an MoE-LoRA architecture to expand model capacity without a substantial increase in inference overhead. This enables different experts to focus on spatial layout for a variety of scenarios, effectively tackling the challenge of scene diversity.

We strategically integrate the MoE-LoRA into the output feed-forward network (FFN) layers of the Flux, while other layers are adapted using standard LoRA for parameter efficiency. Specifically, given an input vector $h$ to the FFN layer, we define the number of experts as $N_e$, the rank of each LoRA as $r$, and the scaling factor as $\alpha$. A lightweight gating network, $g_\theta$, which dynamically routes the input vector $h$ to the most suitable experts, computes a vector of logits, $p \in \mathbb{R}^{N_e}$, for the experts:

$$p = \text{Softmax}(\text{TopK}(W_g \cdot h, k))$$

where $W_g \in \mathbb{R}^{N_e \times d_{in}}$ is the weight of the gating network. $\text{TopK}(\cdot, k)$ enforces sparsity by retaining only the top $k$ logit values and masking the others to $-\infty$, thus activating only a small subset of experts. Each of the $N_e$ experts is an independent LoRA module, parameterized by matrices $W_A^i \in \mathbb{R}^{r \times d_{in}}$ and $W_B^i \in \mathbb{R}^{d_{out} \times r}$. The final output of the MoE-LoRA layer, $h_{out}$, is computed by adding the weighted sum of the selected experts' outputs to the output of the original FFN layer:

$$h_{out} = \text{FFN}(h) + \sum_{i=1}^{N_e} p_i \cdot \left( \frac{\alpha}{r} \cdot W_B^i \cdot W_A^i \cdot h \right).$$

### 4.4 IDENTITY-PRESERVING PREFERENCE OPTIMIZATION

To further enhance the generation quality and align with human preferences, we introduce a post-training stage using reinforcement learning. This final stage aims to refine aesthetic appeal and text-image alignment without compromising the subject fidelity. We adapt the efficient MixGRPO framework (Li et al., 2025b), which confines stochastic optimization to a sliding window $S$. However, standard GRPO with its token-level policy ratios can exhibit instability, particularly when training MoE models due to expert routing fluctuations (Zheng et al., 2025). To mitigate this and better suit our MoE-LoRA architecture from Sec. 4.3, we replace the GRPO objective with the more stable Group Sequence Policy Optimization (GSPO) (Zheng et al., 2025). Specifically, we replace the token-level policy ratio in Eq. (3) with a sequence-level policy ratio $s_i(\theta)$ defined over the denoising steps within the sliding window $S$:

$$s_i(\theta) = \exp\left( \frac{1}{|S|} \sum_{t \in S} \log \frac{\pi_\theta(x_{t+1}|x_t, c, \mathcal{Z})}{\pi_{\theta_{old}}(x_{t+1}|x_t, c, \mathcal{Z})} \right). \tag{7}$$

This sequence-level ratio reflects the overall policy shift for the entire sequence within the optimization window, leading to more stable gradients. The advantage score $A_i$ is calculated as in Eq. (3), but using our composite reward from Eq. (9). The final optimization objective is thus:

$$J_{\text{MixGSPO}}(\theta) = \mathbb{E}_{c, \mathcal{Z}, \{x_i^T\}_{i=0}^N \sim \pi_{\theta_{old}}(\cdot|c, \mathcal{Z})} \left[ \frac{1}{N} \sum_{i=1}^N \min\left( s_i(\theta)A_i, \text{clip}(s_i(\theta), 1-\beta, 1+\beta)A_i \right) \right]. \tag{8}$$

Besides, a good reward model is very important for online reinforcement learning. We construct a reward model based on three dimensions: aesthetics, text alignment, and subject fidelity. The total reward $R(x_i^T, c, \mathcal{Z})$ for a generated image $x_i^T$ given a text prompt $c_{text}$ and a set of reference subject latents $\mathcal{Z}$ is a weighted sum of three scores:

$$R(x_i^T, c, \mathcal{Z}) = w_{text}R_{text} + w_{aes}R_{aes} + w_{id}R_{id}, \tag{9}$$

where $R_{text}$ is the text alignment reward from a pre-trained CLIP model, $R_{aes}$ is an aesthetic reward from a predictor like HPSv2 (Wu et al., 2023), $R_{id}$ is used to evaluate subject fidelity, and $w_{id}, w_{text}, w_{aes}$ are their corresponding weights, . To accurately measure the subject fidelity of

| Methods | Multi Human Genertaion | | | | | Multi Object Generation | | | | Overall |
|---|---|---|---|---|---|---|---|---|---|---|
| | CLIP-T | Face-Sim | DINO-I | CLIP-I | HPS | CLIP-T | DINO-I | CLIP-I | HPS | AVG |
| MS-Diffusion | 0.2498 | 0.0945 | 0.4767 | 0.5801 | 0.2461 | 0.2887 | 0.4002 | 0.6681 | 0.2685 | 0.3636 |
| MIP-Adapter | 0.2631 | 0.2117 | 0.6959 | 0.7140 | 0.2791 | 0.2984 | 0.5470 | 0.7776 | 0.2374 | 0.4471 |
| OmniGen | 0.2741 | 0.3238 | 0.7225 | 0.7642 | 0.2996 | 0.3151 | 0.6727 | 0.8085 | 0.2561 | 0.4930 |
| UNO | 0.2645 | 0.1474 | 0.5972 | 0.6489 | 0.2954 | 0.3259 | 0.7374 | 0.8392 | 0.2676 | 0.4582 |
| OmniGen2 | **0.2837** | 0.2453 | 0.6788 | 0.7205 | **0.3103** | 0.3310 | 0.7538 | 0.8470 | **0.2872** | 0.4953 |
| DreamO | 0.2747 | 0.3345 | 0.7441 | 0.7988 | 0.3056 | 0.3207 | 0.7394 | 0.8393 | 0.2637 | 0.5134 |
| XVerse | 0.2591 | 0.4117 | 0.7665 | 0.8027 | 0.2498 | 0.2981 | 0.7449 | 0.8456 | 0.2595 | 0.5153 |
| Ours | 0.2753 | **0.5284** | **0.8294** | **0.8524** | 0.2915 | **0.3380** | **0.7824** | **0.8608** | 0.2748 | **0.5592** |

Table 1: Quantitative comparison with state-of-the-art methods on the multi-human and multi-object generation. The best results are in **bold**, and the second-best are underlined. Our method outperforms others, especially in subject fidelity metrics, and achieves the highest overall average score.

multi-subject generation results, we built the Multi-ID Alignment Reward using the Hungarian matching algorithm. For human subjects, we first employ a face detector (Deng et al., 2020) to extract facial embeddings from each reference image. We then apply the same detector to the generated image to identify all present faces and extract their embeddings. Then we construct a pairwise similarity matrix $C$ where $C_{ij}$ is the cosine similarity between the embedding of the $i$-th reference face and the $j$-th detected face. The Hungarian algorithm is then used to solve the assignment problem by finding an assignment matrix $X \in \{0, 1\}^{N_{ref} \times N_{gen}}$ that maximizes the total similarity:

$$\max_X \sum_{i=1}^{N_{ref}} \sum_{j=1}^{N_{gen}} C_{ij} X_{ij} \quad \text{s.t.} \sum_{j=1}^{N_{gen}} X_{ij} \leq 1, \sum_{i=1}^{N_{ref}} X_{ij} \leq 1. \tag{10}$$

where $N_{ref}$ and $N_{gen}$ are the number of reference and generated faces, respectively. These ensure each face is matched at most once, preventing reward hacking, stopping the model from using attribute leakage to generate multiple "average faces" for an unearned high reward. For object subjects, each reference object is pre-annotated with a text prompt. We leverage Florence-2 (Xiao et al., 2024) and SAM2 (Ravi et al., 2024) to locate the corresponding object in the generated image. Then we compute the cosine similarity between the DINOv2 (Oquab et al., 2023) embeddings of the segmented region and the reference object. We provide the complete pseudocode at Appendix A.

## 5 EXPERIMENTS

### 5.1 EXPERIMENTAL SETUP

**Implementation Details.** To achieve precise multi-subject customized generation, our training process is divided into three stages: Single-Subject Pre-training, Multi-Subject Customization Training, and Identity-Preserving Preference Optimization. Following (Wu et al., 2025c), we set the resolution of generated images to $512 \times 512$ , the resolution of reference images to $320 \times 320$, and each LoRA module's rank to $r = 512$. For the MoE-LoRA applied to the FFN layers, we configure it with 4 experts and activate 1 expert per forward pass. For reinforcement learning, we configure with a sampling step 16, a window size of $w = 2$, a shift interval of $\tau = 50$, and a window stride of $s = 1$. More details settings can be found in Appendix B.

**Datasets and BenchMark.** We constructed separate datasets for multi-human and multi-object generation. Due to the scarcity of public multi-human customization datasets with adequate annotations, we designed a data collection pipeline to curate a new dataset from videos for multi-human customization. We supplemented the details in the Appendix C. A total of 200k pairs of cross-pair data were obtained for training. For the reinforcement learning stage, we curated a face collection of 80 celebrities and 80 non-celebrities. We paired the faces in the collection and used Qwen2.5-VL (Bai et al., 2025) to generate a prompt for each pair, resulting in 12,720 data points. We randomly selected 1,000 of these data points as our benchmark, and the remaining data was used as reinforcement learning training data. For multi-object customization, we use the public MUSAR-Gen (Guo et al., 2025) Dataset as our foundation. We use Florence-2 and SAM to obtain the detailed positions of reference objects. To ensure segmentation quality, we further employ Qwen2.5-VL to filter the results. We split 1,000 samples from this dataset that were not seen during training as a benchmark.

**Evaluation metrics.** Following the (Le et al., 2025; Mou et al., 2025), we evaluate generated image quality using standard metrics. We calculate the cosine similarity between the prompt and the image

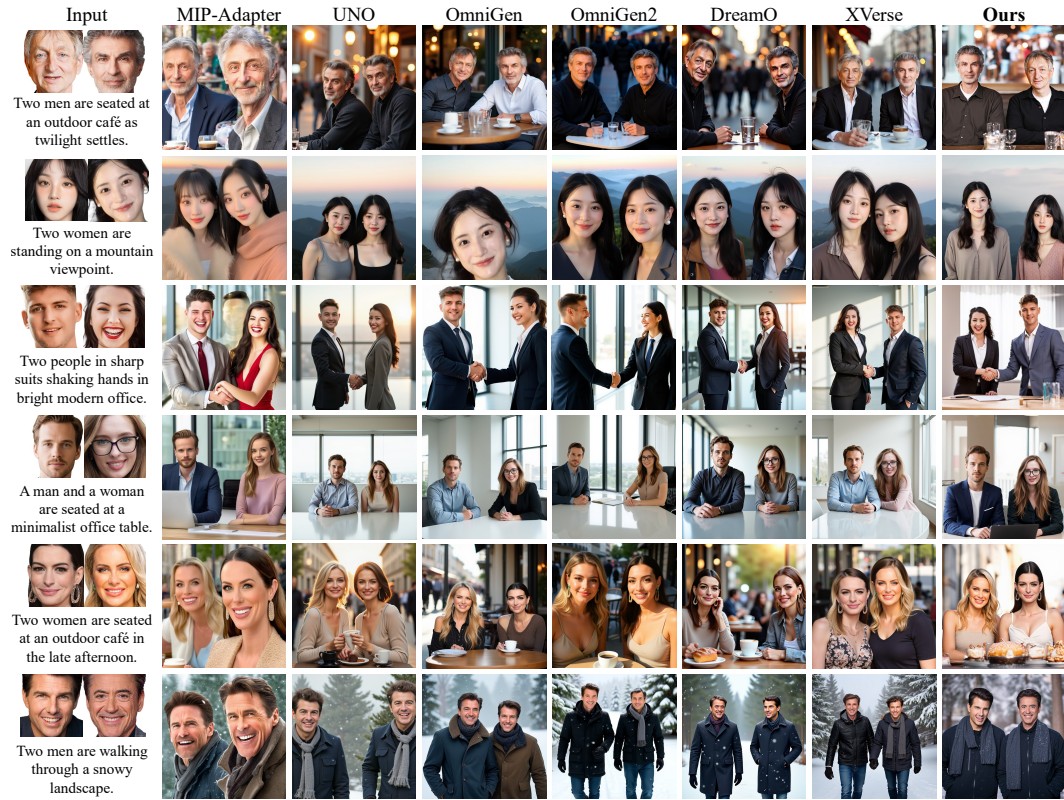

Figure 4: Qualitative comparison with existing methods on the multi-human generation (Zoom in for best visual comparison). Our method significantly improves subject fidelity.

CLIP (Radford et al., 2021) embeddings (CLIP-T) to evaluate text fidelity. We also use HPSv2 for aesthetic and human preference scoring. For subject fidelity, we employ cosine similarity measures between generated images and reference images within CLIP and DINO (Zhang et al., 2022) spaces, referred to as CLIP-I and DINO-I scores, respectively. Note that in order to accurately calculate CLIP-I and DINO-I, we first use a face detector (Deng et al., 2019) or Florence-2 to accurately locate the position of subjects in the generated image, and then calculate them. Additionally, for multi-human generation, we incorporate Face Similarity (Face-Sim) (Deng et al., 2019) and the Hungarian algorithm, enabling them to assess subject fidelity more accurately.

## 5.2 QUANTITATIVE COMPARISON

We conducted comprehensive comparisons with existing methods in both multi-subject and multi-object customized generation. These methods include MS-Diffusion (Wang et al., 2025), MIP-Adapter (Zhong et al., 2025), OmniGen (Xiao et al., 2025), UNO (Wu et al., 2025c), OmniGen2 (Wu et al., 2025b), DreamO (Mou et al., 2025), and XVerse (Chen et al., 2025a). The results, as shown in Tab. 1, demonstrate that our method achieves significant improvements over existing approaches, particularly in terms of subject fidelity. In the multi-human generation task, our model shows a commanding lead in identity preservation, achieving the top scores in subject fidelity. Notably, the significant 28.3% relative improvement in the Face-Sim over the next-best method highlights our model's ability to effectively distinguish between different subjects and preserve their individual detailed features. This is crucial for accurately and reliably accomplishing the highly sensitive and challenging task of multi-human customization. Concurrently, the model remains highly competitive in text-image alignment, achieving a strong balance between subject fidelity and prompt alignment. This outstanding performance extends to the multi-object generation benchmark, where our method again secures the top ranks in text alignment and subject fidelity, validating its robustness and versatility. While HPS score tends to vibrant yet unnatural ("oily") outputs, our model is specifically trained for multi-human customization with an emphasis on photorealism. To preserve this quality, we follow Xue et al. (2025) and employ the CLIP Score during reinforcement

| IDAR | MoE-LoRA | IPPO | CLIP-T | Face-Sim | DINO-I | CLIP-I | HPS |
|------|----------|------|--------|----------|--------|--------|-----|
| ✗ | ✗ | ✗ | 0.2645 | 0.1474 | 0.5972 | 0.6489 | 0.2954 |
| ✓ | ✗ | ✗ | 0.2637 | 0.4983 | 0.7953 | 0.8032 | 0.2653 |
| ✓ | ✓ | ✗ | 0.2674 | 0.5154 | 0.8107 | 0.8480 | 0.2661 |
| ✓ | ✓ | ✓ | 0.2753 | 0.5284 | 0.8294 | 0.8524 | 0.2915 |

Table 2: Ablation study of the core module of MultiCrafter. IDAR is Identity-Disentangled Attention Regularization (Sec. 4.2), MoE-LoRA is our Efficient Adaptive Expert Tuning (Sec. 4.3) , and IPPO is our Identity-Preserving Preference Optimization (Sec. 4.4). The results demonstrate the effectiveness of each module.

learning. Therefore, we only achieve competitive objective scoring results in aesthetics, but we can generate more realistic results. Ultimately, by achieving the best overall score, our method achieves significant improvements over existing methods, validating the effectiveness of our framework.

## 5.3 QUALITATIVE COMPARISON

We provide a qualitative comparison against existing methods on the more challenging task of multi-human customized generation in Fig. 4. As shown in the first two rows of Fig. 4, methods trained directly with In-Context Learning, such as UNO, OmniGen, and OmniGen2, struggle with attribute confusion when generating subjects of the same gender, which degrades subject fidelity. In contrast, our method accurately preserves the unique features of each individual. Our method maintains strong subject fidelity even in interactive scenarios, as shown in the third row of Fig. 4. This result validates the effectiveness of our framework. Aesthetically, while DreamO and OmniGen2 produce vibrantly colored images, our approach generates images with a higher degree of realism. Additional qualitative comparisons for both multi-subject generation (includes both human and object) are provided in the Appendices G to I.

## 5.4 ABLATION STUDIES

We conduct detailed ablation studies to validate the effectiveness of each component of our proposed framework. Since the multi-human customization evaluation metrics is more accurate, we conducted an ablation experiment on multi-human customization. We use UNO as our baseline for comparison, and the detailed results are presented in Tab. 2. The baseline model relies solely on a simple reconstruction loss. As shown in Tab. 2, the base model performs poorly in terms of subject fidelity, achieving a Face-Sim score of only 0.1474. This confirms that a simple objective is insufficient for handling complex multi-subject scenarios. Upon introducing our Identity-Disentangled Attention Regularization (IDAR), we observe a dramatic improvement across all subject fidelity metrics. As shown on the left of Fig. 2, our method successfully separates the attention of different subjects. This result underscores the critical role of our IDAR in explicitly disentangling subject features and preventing attribute leakage. We attribute the slight decrease in the HPS score to the quality of our training dataset, a point we discuss further in the Appendix E. Building on this, we integrate the MoE-LoRA architecture, which yields further gains across subject fidelity while also improving text alignment. This demonstrates that our strategy of using MoE to allow different experts to specialize in handling diverse spatial layouts is effective in enhancing generation quality. Finally, Identity-Preserving Preference Optimization (IPPO) further boosts subject fidelity. More importantly, it significantly enhances alignment with human preferences, leading to notable improvements in both text alignment and aesthetic scores. This shows that our IPPO successfully aligns the model's output with human preferences without compromising the subject fidelity.

## 6 CONCLUSION

In this paper, we introduced MultiCrafter, a novel framework designed to address the critical challenges of feature bleed and identity degradation in multi-subject customized generation. Our framework uses Identity-Disentangled Attention Regularization to prevent "attention bleeding" and alleviate the degradation of subject fidelity caused by attribute leakage. Then, we introduce a Mixture-of-Experts architecture to enhance model capacity. We further align the model with human preferences using a novel online reinforcement learning framework featuring a Multi-ID Alignment Reward and the stable GSPO algorithm. Experiments show that MultiCrafter significantly improves subject fidelity while better aligning with human preferences and generating realistic images.

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

# A    MORE DETAILS FOR IDENTITY-PRESERVING PREFERENCE OPTIMIZATION

We provide a detailed algorithm execution flow for Identity-Preserving Preference Optimization Sec. 4.4, as shown in Algorithm 1.

---

**Algorithm 1** Identity-Preserving Preference Optimization Training Process

---

**Require:** initial policy model $\pi_\theta$; composite reward model $R$; prompt dataset $\mathcal{C}$; reference subjects dataset $\mathcal{Z}_{\text{data}}$; total sampling steps $T$; number of samples per prompt $N$; sliding window $W(l)$, window size $w$, shift interval $\tau$, window stride $s$
1: Init left boundary of $W(l)$: $l \leftarrow 0$
2: **for** training iteration $m = 1$ **to** $M$ **do**
3:      Sample batch prompts $\mathcal{C}_b \sim \mathcal{C}$ and corresponding subjects $\mathcal{Z}_b \sim \mathcal{Z}_{\text{data}}$
4:      Update old policy model: $\pi_{\theta_{\text{old}}} \leftarrow \pi_\theta$
5:      **for** each prompt $c \in \mathcal{C}_b$ and subject $\mathcal{Z} \in \mathcal{Z}_b$ **do**
6:          Init the same noise $s_0 \sim \mathcal{N}(0, \mathbf{I})$
7:          **for** generate $i$-th image from $i = 1$ **to** N **do**
8:              **for** sampling timestep $t = 0$ **to** $T - 1$ **do**              ▷ $\pi_{\theta_{\text{old}}}$ mixed sampling loop
9:                  **if** $t \in W(l)$ **then**
10:                      Use SDE Sampling to get $s_{t+1}^i$
11:                  **else**
12:                      Use ODE Sampling to get $s_{t+1}^i$
13:                  **end if**
14:              **end for**
15:          **end for**
16:          Calculate advantage: $A_i \leftarrow \frac{R(s_T^i, c, \mathcal{Z}) - \text{mean}(\{R(s_T^j, c, \mathcal{Z})\}_{j=1}^N)}{\text{std}(\{R(s_T^j, c, \mathcal{Z})\}_{j=1}^N)}$
17:          **for** optimization timestep $t \in W(l)$ **do**              ▷ optimize policy model $\pi_\theta$
18:              Update policy model via gradient ascent: $\theta \leftarrow \theta + \eta \nabla_\theta \mathcal{J}_{GSPO}$
19:          **end for**
20:      **end for**
21:      **if** $m \bmod \tau = 0$ **then**              ▷ move sliding window
22:          $l \leftarrow \min(l + s, \, T - w)$
23:      **end if**
24: **end for**

---

# B    MORE IMPLEMENTATION DETAILS.

In this section, we provide more details on the hyperparameter settings and specific training details. As mentioned in the main text, we divide the training phase into three parts, and the details of each phase are as follows: *1. Single-Subject Pre-training.* We first pre-train the model for 40,000 steps on an internal single-subject dataset to equip it with foundational subject customization capabilities. In this stage, only $\mathcal{L}_{diff}$ is used for supervision. We use the AdamW optimizer with a $3 \times 10^{-5}$ learning rate and a weight decay of $1 \times 10^{-2}$. We use 8 cards for training and set the batch size of each card to 6. *2. Multi-Subject Customization Training.* Then we train two models for customized human generation and object generation, respectively. For multi-human customized generation, we decrease the learning rate to $1 \times 10^{-5}$, set the loss weight $\lambda = 0.3$, and introduce the attention loss $\mathcal{L}_{attn}$. This stage runs for 25,000 steps. We use 8 cards at this stage and set the batch size to 4. For multi-object customized generation, since the size of the dataset is smaller than multi-human datasets, we only train for 15,000 steps. *3. Identity-Preserving Preference Optimization.* Finally, we fine-tune the model using our proposed reinforcement learning method. Following (Li et al., 2025b), we configure with a sampling step of 16, a window size of $w = 2$, a shift interval of $\tau = 50$, and a window stride of $s = 1$. This stage consists of 300 steps. For multi-human customization, the reward weights are set to $w_{id} = 0.5$, $w_{text} = 1.4$, and $w_{aes} = 0.7$. For multi-object customization, we adjust the subject fidelity weight to $w_{id} = 1.0$, while keeping $w_{text} = 1.4$ and $w_{aes} = 0.7$. In this stage, we used 16 cards for training and set the batch size of each card to 1.

## C    TRAINING DATASET CONSTRUCTION PIPELINE.

Due to the scarcity of public multi-human customization datasets with adequate annotations, we designed a comprehensive and automated data curation pipeline as shown in Fig. 5. This pipeline processes raw video clips to generate structured training samples, each containing a target image with multiple subjects, corresponding identity reference images, segmentation masks, and a detailed textual description. The entire workflow ensures subject fidelity, high image quality, and rich annotation. Specifically, we sample video clips featuring two individuals from a large database. The process begins with frame selection and subject localization. For each input video, we sample an initial frame as the source for reference images and a middle frame as the target scene. We first employ a YOLO-Pose (Maji et al., 2022) to obtain initial bounding boxes and keypoint information for each person. Following localization, we leverage the Segment Anything Model (SAM) (Ravi et al., 2024) to generate high-fidelity segmentation masks for each individual, effectively isolating them from the background. To refine this output, only the largest connected component of the mask is retained. A critical subsequent step is ensuring subject fidelity across frames. We apply a face detection modell (Deng et al., 2019) to the segmented portraits to locate facial regions, and then use a face recognition model (Deng et al., 2019) to extract a normalized feature embedding for each face. By computing the cosine similarity between embeddings from the reference and target frames, we enforce a stringent threshold to discard pairs where identity cannot be confidently verified. Once a pair of frames passes this verification, the pipeline generates the final training sample. The segmented portraits and cropped faces from the initial frame are saved as the reference =images. The target frame is cropped into a square as target image, with its corresponding body and face masks preserved. Finally, a powerful vision-language model, Qwen2.5-VL (Bai et al., 2025), is prompted with the reference images and the target image to produce a rich text prompt of the entire scene, ensuring descriptive consistency for each subject.

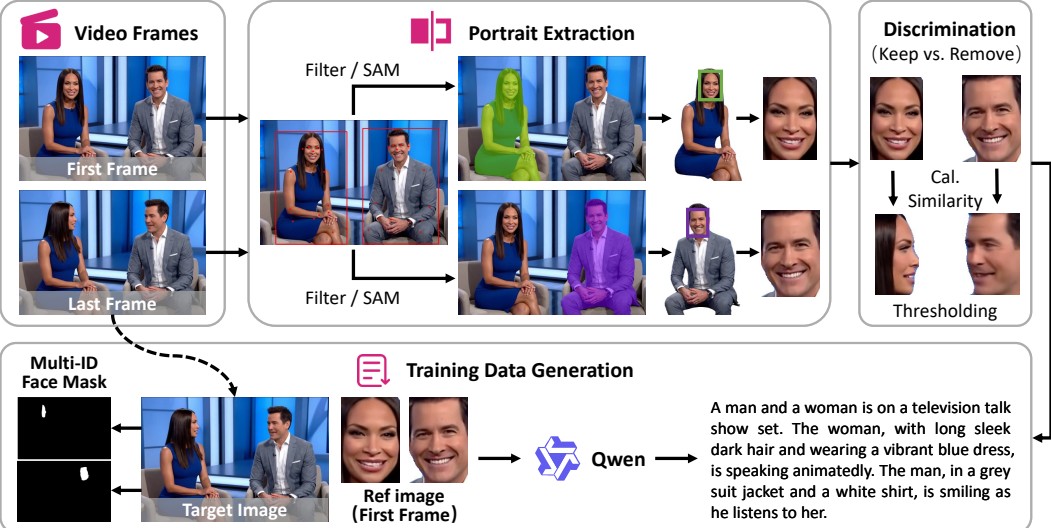

Figure 5: Data processing pipeline for customized multi human image generation.

## D    MORE DETAILS FOR OUR BENCHMARK.

To advance research on high-fidelity multi-subject generation, we constructed a benchmark dataset by collecting images from publicly available sources and extracting the corresponding facial regions as reference images. The dataset comprises 80 celebrities and 80 non-celebrities, covering diverse attributes in terms of gender, age, and ethnicity (male/female; young/elderly; Caucasian, Black, and Asian). We used this face collection as a reference pool and paired faces within it. For each pair, we employed Qwen2.5-VL to generate distinctive natural-language prompts to provide diverse textual descriptions. Representative samples are shown in Fig. 6.

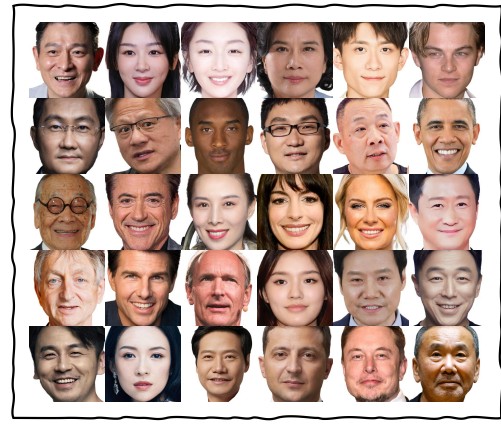 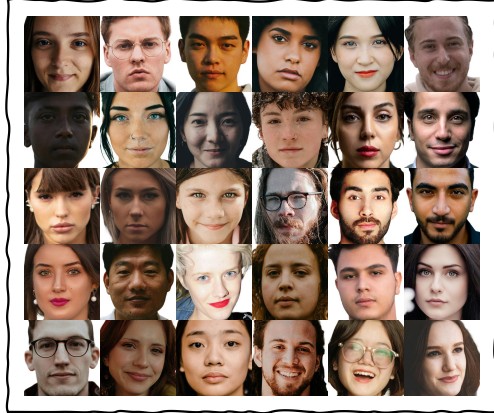

celebrity

non-celebrity

Figure 6: Visualization for part of our multi-human evaluation benchmarks.

# E LIMITATION.

Although MultiCrafter has achieved excellent performance in multi-subject driven image generation tasks, our work still has certain limitations, which also point the way for future research.

First, the scale and quality of the training data are the primary limiting factors. Currently, high-quality, publicly available datasets for multi-subject driven generation remain scarce (Wu et al., 2025c; Chen et al., 2025a). Although we have designed a complete automated data processing pipeline to extract training samples from videos, our dataset is still limited in scale and diversity due to the quantity and quality of open-source video data.

Second, the effectiveness of our method has been primarily validated in two-subject scenarios.Since our multi-person dataset and the public MUSAR dataset (Guo et al., 2025) mainly contain two subjects, the experiments in this paper were centered around this setting. The model's generation capability when handling three or more subjects has not been fully verified. It is worth noting that our framework was designed with scalability in mind; both the attention regularization mechanism and the Multi-ID Alignment Reward (based on the Hungarian algorithm) in the reinforcement learning framework can be directly extended to scenarios with more subjects.

For future work, we plan to explore improvements from both data and model perspectives. On one hand, we will attempt to construct larger, higher-quality datasets containing a more diverse number of subjects by combining synthetic data with image editing Wu et al. (2025a) techniques. On the other hand, we will train and evaluate the model in scenarios with more subjects to further enhance the generalization and robustness of MultiCrafter, enabling it to handle more complex personalized image generation.

# F THE USE OF LARGE LANGUAGE MODELS.

In this paper, we only use the large language model to help polish our text. The large language model has no role in the research conception.

# G MORE RESULTS OF MULTI-HUMAN PERSONALIZATION.

To further demonstrate our method's performance in multi-human personalization, we present qualitative comparisons in Fig. 7 and Fig. 8. The results show that our model effectively preserves the identity of each subject and avoids the "attribute leakage" common in other methods. This outcome validates the efficacy of our Identity-Disentangled Attention Regularization (IDAR). While some baselines produce more stylized outputs that may yield higher HPSv2 scores, this is often at the expense of subject fidelity. Our method prioritizes photorealism and faithful subject appearance consistency, which leads to more reliable results in multi-subject customization.

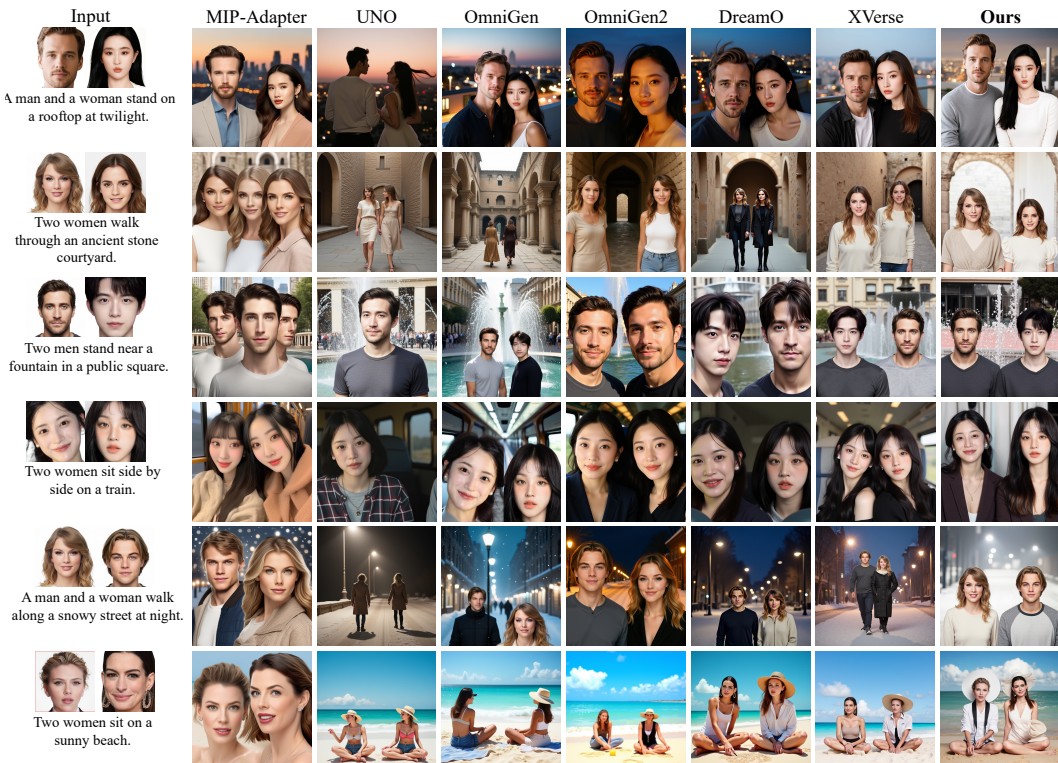

Figure 7: More Visualization of our method in Multi-Human Personalization.

## H  MORE RESULTS OF MULTI-OBJECT PERSONALIZATION.

To evaluate the generalization of our framework, we showcase multi-object customization comparisons in Fig. 9. Our method demonstrates high object fidelity, accurately preserving core visual attributes such as a toy's texture or a glass's geometry. In contrast, competing approaches often introduce artifacts like deformation and detail loss. This highlights our model's strength in precise subject representation rather than hyper-stylization, a crucial capability for practical applications that require accuracy.

## I  MORE RESULTS OF SINGLE-SUBJECT PERSONALIZATION.

Effective multi-subject generation builds on strong single-subject performance. We validate this capability in Fig. 10 and Fig. 11, showing six diverse samples for each of four individuals and six frontal single-subject comparisons against SOTA models. Our method consistently preserves identity across varying styles, poses, and scenes, and even improves fidelity to the reference over baselines. These results confirm that the proposed framework not only enables reliable multi-subject generation but also enhances single-subject identity fidelity.

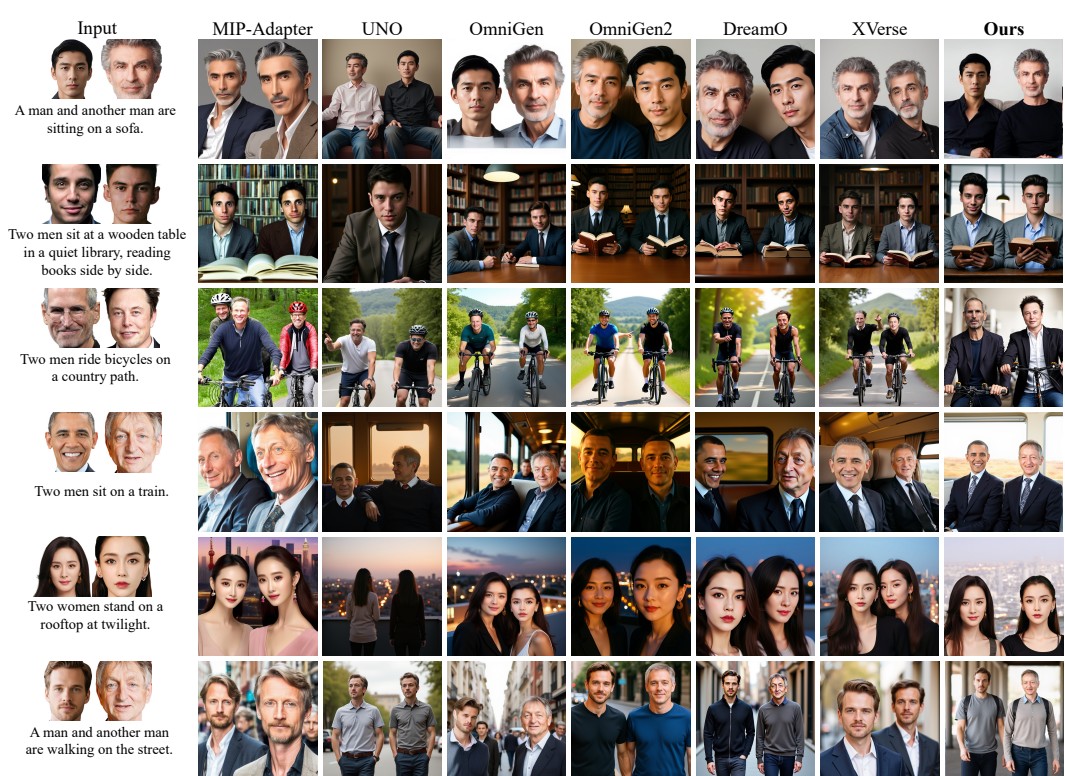

Figure 8: More Visualization of our method in Multi-Human Personalization.

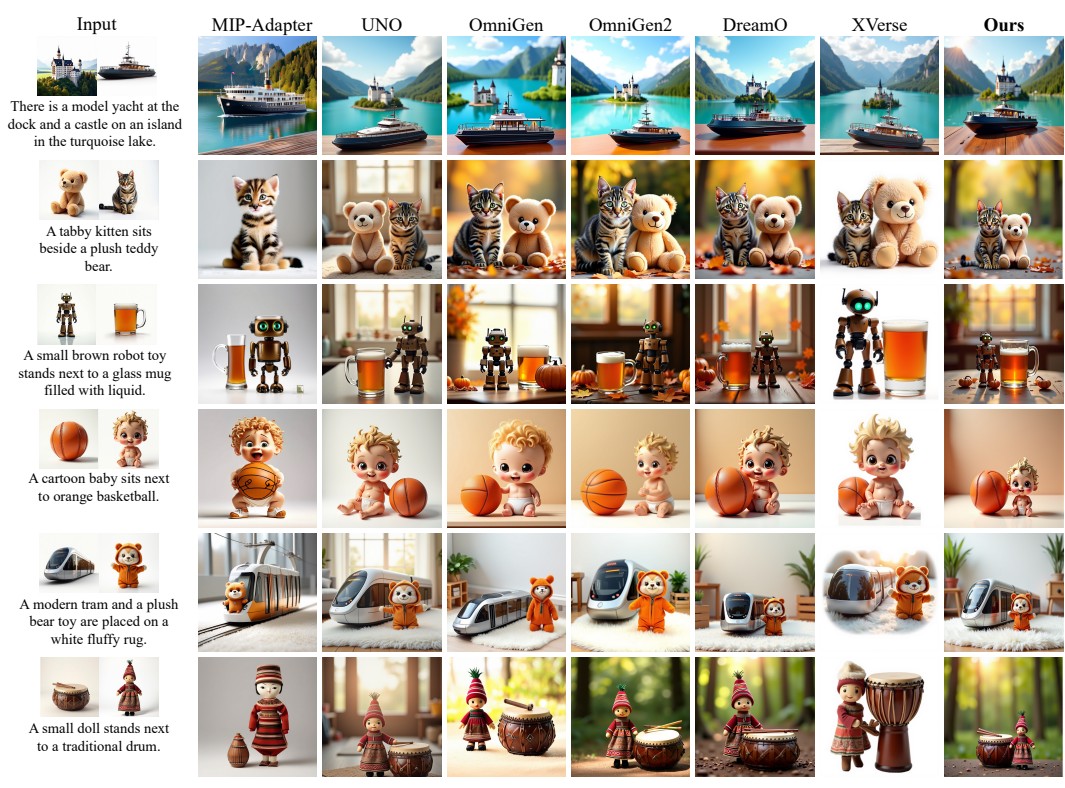

Figure 9: Visualization of our method in Multi-Object Personalization.

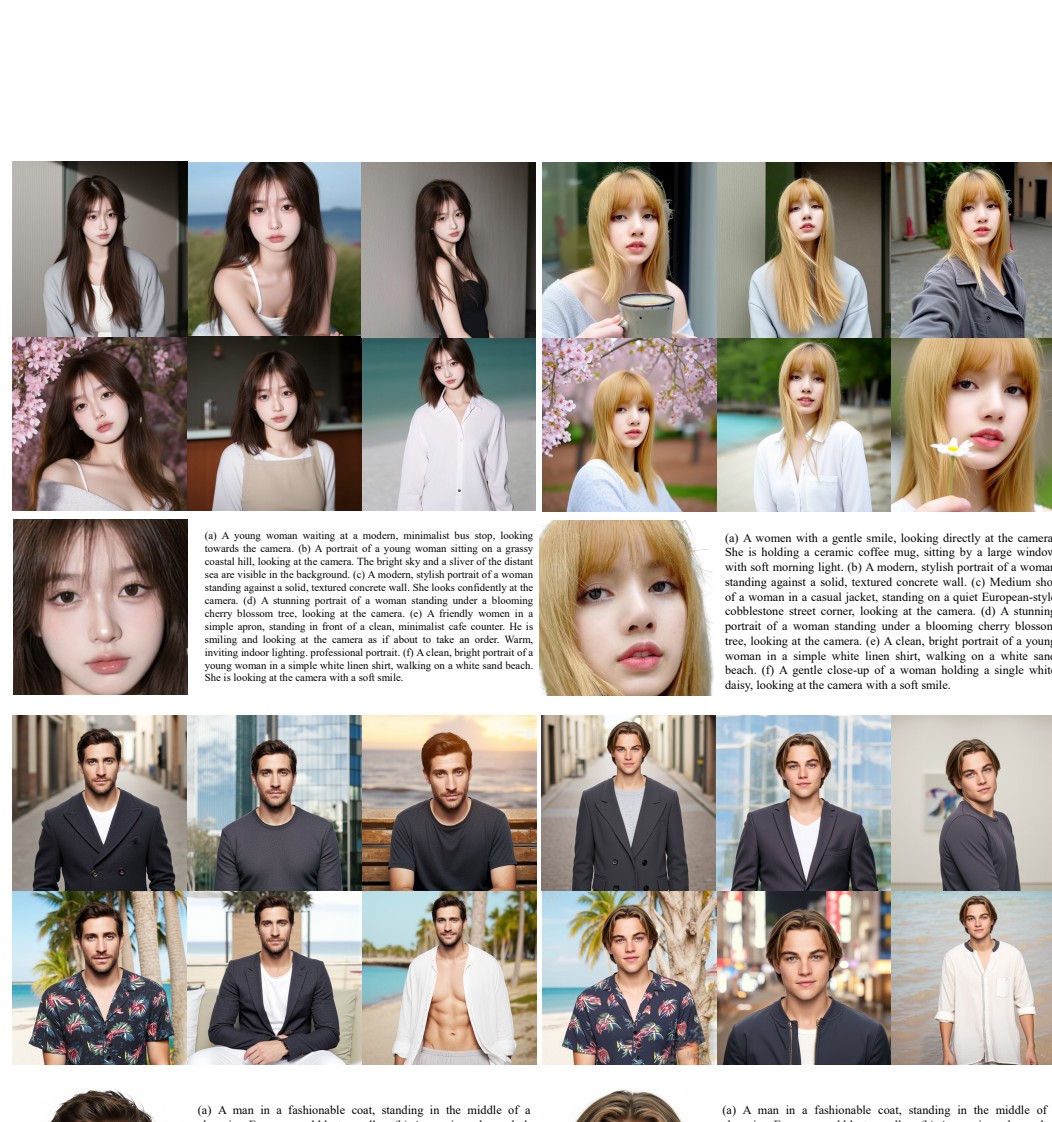

Figure 10: Visualization of our method in Single-Subject Personalization.

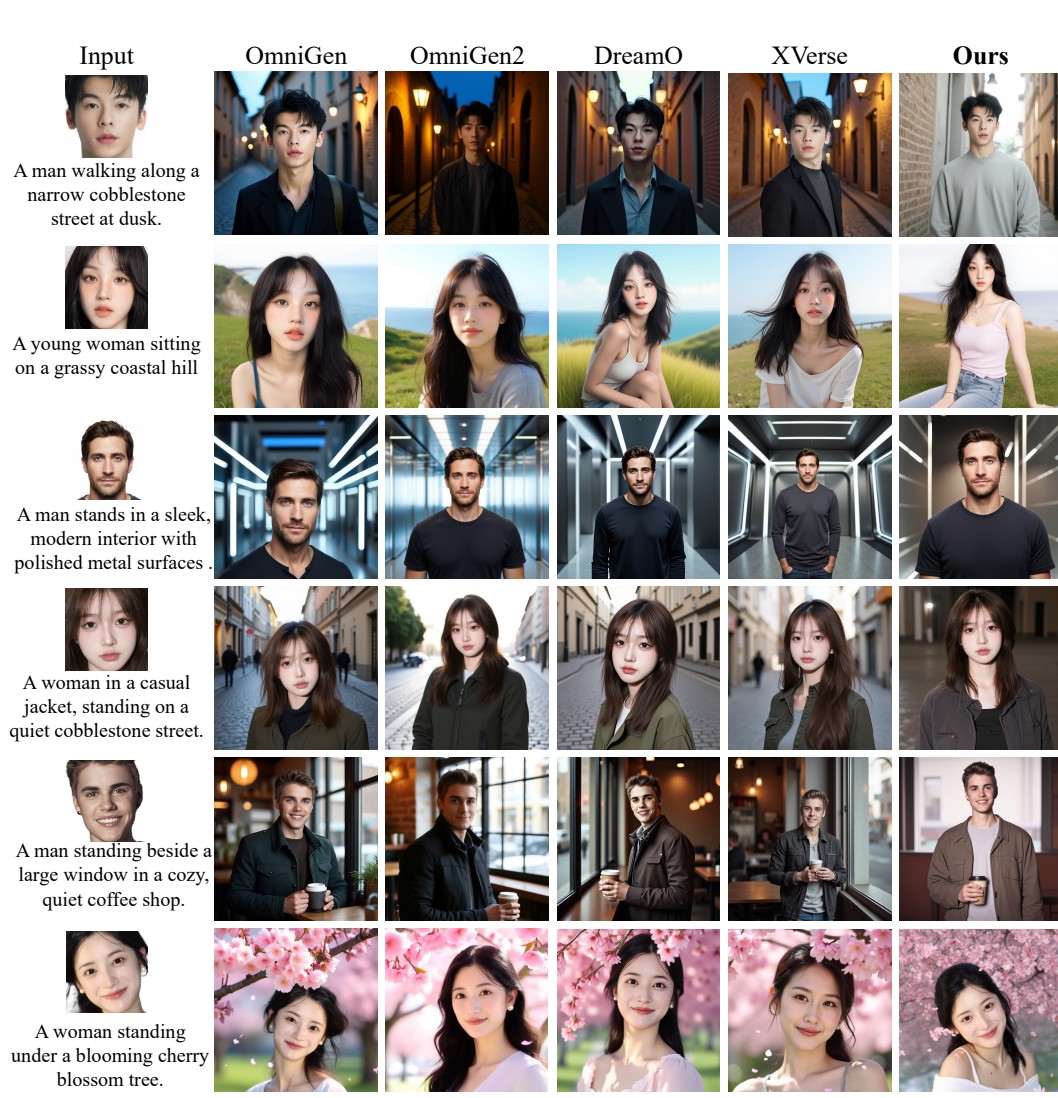

Figure 11: Qualitative comparison with existing methods on the single human generation.

