# OpenReview forum: "MultiCrafter: High-Fidelity Multi-Subject Generation via Spatially Disentangled Attention and Identity-Aware Reinforcement Learning"
_ICLR.cc/2026/Conference — ICLR 2026 Conference Withdrawn Submission_

### Official Review · Reviewer_WvAe · 2025-10-28

**Soundness:** 3
**Presentation:** 3
**Contribution:** 2
**Rating:** 4
**Confidence:** 4

**Summary:**

The work introduces MultiCrafter, a framework designed to generate images that feature multiple personalized subjects within one scene while maintaining each subject’s identity and natural appearance. The focus is on addressing the common issue of attribute leakage, where visual features from one subject unintentionally blend into another during generation.

To overcome this, the authors propose three key components. First, Identity-Disentangled Attention Regularization (IDAR) adds positional supervision during training, forcing the model to separate attention regions for different subjects. Second, Mixture-of-Experts LoRA (MoE-LoRA) expands the model’s capacity by assigning different expert modules to handle diverse spatial layouts and subject combinations without increasing inference cost. Finally, Identity-Preserving Preference Optimization (IPPO) applies an online reinforcement learning stage that aligns generated results with human aesthetic and textual preferences while preserving subject fidelity.

**Strengths:**

This work presents a complete training pipeline for multi-subject generation, covering both architectural and optimization aspects. The overall design feels systematic, from attention disentanglement to expert-based adaptation and preference alignment, forming a coherent and well-structured framework. The experiments are well-organized, and the results show solid improvements in subject fidelity and prompt alignment.

I also find the discussion on the instability between MixGRPO and MoE training quite interesting. The observation that token-level policy updates can become unstable due to expert routing fluctuations is something I hadn’t considered before, it’s a thoughtful and technically grounded point. That said, it would be even stronger if the authors provided some experimental analysis (e.g., training curves or comparison with MixGRPO-only training) to support this claim, as it currently reads more as a theoretical justification.

**Weaknesses:**

While the paper forms a complete pipeline, it also feels somewhat difficult to pinpoint a clear central contribution. The overall design combines several existing components: disentangled attention, MoE tuning, and post-training preference optimization, which makes the framework look more like an accumulation of improvements rather than a single focused idea.

Another concern is the level of manual design involved. Elements such as the attention masks and positional supervision, while effective, introduce heavy handcrafted components that may reduce the model’s generalization ability. In some generated samples, the facial structures appear almost copy-and-paste from the reference images, suggesting limited diversity. I would like to see more results demonstrating variation, for instance, when the reference image is a side face but the generated image shows a different viewpoint or expression.

Lastly, I’m not entirely sure whether extracting only VAE features from the reference images might contribute to this issue. Some discussion or ablation on this aspect would make the work more convincing.

**Questions:**

See weaknesses.

---

### Official Review · Reviewer_nXob · 2025-10-30

**Soundness:** 3
**Presentation:** 2
**Contribution:** 3
**Rating:** 4
**Confidence:** 4

**Summary:**

This paper introduces "MultiCrafter," a novel framework for multi-subject image generation. The authors aim to address two critical limitations in existing methods: 1) severe attribute leakage, which compromises the fidelity of individual subjects, and 2) poor alignment with human aesthetic preferences. To solve this, the proposed framework makes three key contributions: Explicit Positional Supervision, Mixture-of-Experts (MoE) Architecture and Online Reinforcement Learning (RL) for Alignment. Experiments confirm that MultiCrafter significantly improves subject fidelity while producing results that are better aligned with human preferences.

**Strengths:**

1. Well-specified method components: each key idea—explicit positional supervision, Mixture-of-Experts capacity expansion, and the online RL alignment mechanism—is described with enough implementation detail to understand how it would be applied in practice.
2. Strong visual evidence: attention visualizations and side-by-side qualitative comparisons clearly demonstrate how positional supervision reduces attribute leakage, making the improvements intuitive and compelling.
3. Comprehensive and targeted ablations: the ablation suite systematically isolates the contributions of positional supervision, MoE and the identity preserving preference optimization strategies, providing convincing causal evidence for each component’s benefit.

**Weaknesses:**

1. The MoE contribution in the paper is relatively incremental; as a task-agnostic, general-purpose module, its necessity and motivation for being introduced specifically in the multi-object generation task should be further justified both conceptually and empirically.
2. Attention disentanglement is not a novel idea in multi-object customized generation tasks[i, ii, iii]; the distinctions and connections between this approach and similar methods should be clarified in the related work and experiments.
3. Considering that aesthetic quality and realism in visual evaluation are hard to capture with objective metrics, a user study (subjective evaluation) is necessary to compare the proposed method against other methods and the ablation variants.
[i] Fastcomposer: Tuning-free multi-subject image generation with localized attention. IJCV.
[ii] Customizable image synthesis with multiple subjects. Neurips 2023.
[iii] DisenStudio: Customized Multi-subject Text-to-Video Generation with Disentangled Spatial Control. MM 2024.

**Questions:**

For the Mixture-of-Experts (MoE) part, it would be helpful to provide some routing statistics or visualizations to help readers better understand how the experts are utilized.

---

### Official Review · Reviewer_FX84 · 2025-11-01

**Soundness:** 2
**Presentation:** 2
**Contribution:** 2
**Rating:** 4
**Confidence:** 4

**Summary:**

The paper proposes MultiCrafter, a framework for high-fidelity, preference-aligned multi-subject image generation. The work focuses on addressing attribute leakage and attention entanglement that commonly occur in existing in-context-learning concepts.

The core contributions are:
1. Identity-Disentangled Attention Regularization (IDAR): an explicit positional supervision mechanism to disentangle attention across subjects and prevent attribute leakage.
2. Efficient Adaptive Expert Tuning (MoE-LoRA): a Mixture-of-Experts adaptation that increases model capacity and diversity handling without increasing inference cost.
3. Identity-Preserving Preference Optimization (IPPO): an online reinforcement learning framework with Multi-ID Alignment Reward and GSPO to align outputs with human aesthetic and semantic preferences.

Experiments and evaluations on multi-human and multi-subject benchmarks are established to prove the effectiveness of the proposed method, yet not totally comprehensive.

**Strengths:**

+ The authors figure out the existing problem in current ICL in terms of applying to multi-subject generation and propose the solution corresponding to each problem with evidence.
+ The novelty lies in the approach of addressing multi-subject generation despite still relying on combining several existing methods. The proposed framework addresses a key challenge for personalization in text-to-image models, scaling from single- to multi-subject fidelity while preserving user identities.

**Weaknesses:**

- The proposed MoE-LoRA, GSPO, and also IPPO contributions are trivial, while the framework is the integration of those components. I consider the proposals lack novelty when utilizing the existing work.
- The evaluation of the proposed methods lacks comprehension. In this generation case, qualitative evaluation should be the combination of evaluation metrics and user studies. It is recommended to have a user studies report on the comparison among SoTA methods (in Figure 4) to fairly compare the results of the proposed methods.
- The paper claims to address multi-subject generation tasks. However, only two-subject cases are evaluated, “three or more subjects has not been fully verified”. It is better to consider and give a brief look at some more-than-two-subject cases to strengthen the claim. The recommended datasets should be CelebA-HQ, COCO-person, etc.
- The authors report that while realism improves, aesthetic scores (HPS) are only “competitive”. A more explicit discussion on this trade-off would be necessary to better understand the effectiveness of the proposed method.
- Please provide more empirical evidence to support the claim that MoE-LoRA is introduced “without a substantial increase in inference overhead”.
- Mics: Please proofread for typos, e.g., “in-cotext-learning”

**Questions:**

Please see the above comments.

**Details Of Ethics Concerns:**

N.A.

---

### Official Review · Reviewer_UZvN · 2025-11-02

**Soundness:** 3
**Presentation:** 3
**Contribution:** 2
**Rating:** 6
**Confidence:** 4

**Summary:**

This paper proposes the MultiCrafter framework aimed at addressing the issue of attribute leakage and insufficient alignment of human preferences in existing In-Context-Learning (ICL) paradigm for multi-subject image generation. It prevents attention entanglement by identity-disentanglement attention regularization to reduce attribute leakage and enhance subject fidelity, integrates MoE-LoRA architecture to expand model capacity, and designs an online reinforcement learning framework with Multi-ID Alignment Reward and GSPO to align human preferences. The evaluation experiments are conducted on a self-built multi-human dataset and the multi-object MUSAR-Gen dataset.

**Strengths:**

1. In response to the problem of subject fidelity degradation in existing ICL-based methods in multi-subject setting, this paper clearly points out the key point of "attribute leakage caused by attention entanglement" in multi-subject image generation, and proposes targeted identity-disentanglement attention regularization. By applying explicit positional supervision to the double blocks in the model, the problem of attribute leakage is effectively alleviated from the root of attention mechanism.
2. The quantitative experimental results indicate that MultiCrafter has achieved significant improvements compared to existing methods, especially in terms of subject fidelity. The quantitative comparison results presented in the paper are also promising.

**Weaknesses:**

Major Weaknesses:
Several key points warrant discussion:
1. The authors mention in section 4.2 that explicit positional supervision in identity disentanglement attention regularization requires pre-annotating the ground-truth mask corresponding to the subject within the generated image in the training data. However, there is no explanation on how to complete inference without the ground-truth mask.
2. The authors claim in section 4.3: MoE-LoRA enables different experts to focus on spatial layout for a variety of scenarios. However, in the paper, there is no theoretical or experimental evidence to support this claim.
3. According to Appendix B, MultiCrafter is pre-trained on an internal single-subject dataset. However, the paper lacks a brief explanation of the training process for other comparative methods. Are other comparison methods also pre-trained and tuned on the same datasets? If not, is the internal pre-training data one of the key factors affecting final performance?

Minor Weaknesses:
1. The title in openreview is inconsistent with the title in the PDF: "Reinforcement Learning" vs "Preference Alignment"
2. Typo: There is an extra comma after the word “weights” in the last line of page 6.

**Questions:**

Please refer to the "Several key points warrant discussion" in the weakness scetion.

---

### Note · Authors · 2025-11-14

**Comment:**

We sincerely thank the reviewers for their diligent work and valuable suggestions. We wish you all the best in your work.

**Withdrawal Confirmation:**

I have read and agree with the venue's withdrawal policy on behalf of myself and my co-authors.